# Endophytes in Agriculture: Potential to Improve Yields and Tolerances of Agricultural Crops

**DOI:** 10.3390/microorganisms11051276

**Published:** 2023-05-12

**Authors:** Declan Watts, Enzo A. Palombo, Alex Jaimes Castillo, Bita Zaferanloo

**Affiliations:** Department of Chemistry and Biotechnology, School of Science, Computing and Engineering Technologies, Swinburne University of Technology, Hawthorn, VIC 3122, Australia; declanwatts@swin.edu.au (D.W.);

**Keywords:** endophytes, plant health, sustainable agriculture, stress tolerance, bioprospecting, secondary metabolites

## Abstract

Endophytic fungi and bacteria live asymptomatically within plant tissues. In recent decades, research on endophytes has revealed that their significant role in promoting plants as endophytes has been shown to enhance nutrient uptake, stress tolerance, and disease resistance in the host plants, resulting in improved crop yields. Evidence shows that endophytes can provide improved tolerances to salinity, moisture, and drought conditions, highlighting the capacity to farm them in marginal land with the use of endophyte-based strategies. Furthermore, endophytes offer a sustainable alternative to traditional agricultural practices, reducing the need for synthetic fertilizers and pesticides, and in turn reducing the risks associated with chemical treatments. In this review, we summarise the current knowledge on endophytes in agriculture, highlighting their potential as a sustainable solution for improving crop productivity and general plant health. This review outlines key nutrient, environmental, and biotic stressors, providing examples of endophytes mitigating the effects of stress. We also discuss the challenges associated with the use of endophytes in agriculture and the need for further research to fully realise their potential.

## 1. Introduction

Mutualistic symbiotic endophyte-plant relationships are surmised to be ubiquitous amongst higher plants; every plant on earth contains one or more endophytic fungal (EF) or bacterial (EB) species residing within the leaf, stem, roots, and flower or fruit of plants [1,2]. These relationships have been well documented to commonly give rise to specialised characteristics within plants as a result of the endophyte presence [3]. These properties range from an improved defence against disease and pathogens—antimicrobial, anticancer, antioxidant, anti-inflammatory, insecticidal—to the promotion of plant growth through improved nutrient acquisition and stress tolerance [4].

The relationship between plants and endophytes is one ranging from symbiosis to light pathogenicity. Typically, the plant provides carbohydrates that are vital for endophyte growth, and in return the endophytes’ secondary metabolites can be provided to the plants with tolerances to a breadth of stressors such as drought, increased salinity, nutrient imitation, etc. [1]. As described by Schulz and Boyle [5], 51% of the novel bioactive compounds discovered in-planta were of endophytic origin. This potential pool of bioactive compounds results in the ability to identify and naturally produce composites with medicinal, pharmaceutical, and agricultural applications on a large-scale [6].

Society currently stands at risk of increasing our antimicrobial resistance to bacteria across a range of infections, including bacteria involved in human diseases and plant pathogens causing food spoilage. Both risks weigh upon the fundamental aspects of our society, being healthcare and food supply, which in turn compounds the impact of an ever-increasing population [7,8]. The discovery of novel bioactive secondary metabolites of endophytic origin offers a new direction to combat pathogens. The bioactive metabolites that are produced are considered secondary due to the speculation that they are not essential for the organisms’ growth or reproduction, and instead are a result of evolution in order to protect its source of shelter and nutrition, the host [9].

Endophyte-plant relationships are complex and, only in the last few decades, has the web begun to untangle, with positive characteristics being reported in individualised endophyte-plant interactions. Often, reports are heavily based in favour of the beneficial characteristics, as expected, where confirmational evidence is found. However, negative impacts are scarcely reported or investigated, leaving space for scepticism amongst the theories for large scale applications. As the knowledge base of endophyte-plant relationships expands, it is important to conduct investigations with due diligence in both the positive and negative effects of endophytes, to gain a true understanding of the capabilities of their applications.

## 2. Requirements for Increased Crop Productivity: Climate Change and Population Growth

The population is expanding at a rate that poses a threat to the current capacity of agricultural practices and improvements to crop yields is required to sustain the continuous growth. The global population is predicted to top 9.7 billion in 2050 and 10.4 billion by 2080 [10], not only increasing the total produce yield required to sustain said population, but also reducing the current farmable land in consequence of an increased demand for housing land. As outlined in Figure 1, providing residential land (grey) from limited farmland(green) provokes a need to improve crop tolerances to colonise previously unsuitable/marginal land (orange). Particularly in Australia, farmable land is limited by the water availability and the soil salt concentration, therefore, increasing the ability of plants—with endophytes—to grow at lower moisture or higher salt concentrations unlocks this land for agricultural uses [11,12,13].

The impacts of this increased demand of residential land—if left unsolved—further compound problems of starvation across Asia and Africa (Figure 2). Alongside starvation rates increasing to 9.9% (globally) in 2020, up from 8.4% in 2019, a gradual rise of undernutrition, consequence of the covid-19 pandemic and its effects on global food supply, is projected [14]. It is vitally important that starvation trends take a downward trend if the projected population figures are to come true. Failing to address the current starvation and undernutrition could exponentially increase food supply chain disruptions, impacting low-socioeconomic/third-world populations significantly. Although dire situations such as that in Africa require major interventions in irrigation and fertilisation, endophyte-enhanced plants can still play a role in the biofortification of crop plants.

Nutritional crop limitations, with regards to essential dietary nutrients, are also associated to undernutrition. Therefore, it is important to understand the properties of agricultural crops in relation to the nutritional limitations to nourish troubled nations efficiently and effectively [15].

**Figure 2 microorganisms-11-01276-f002:**
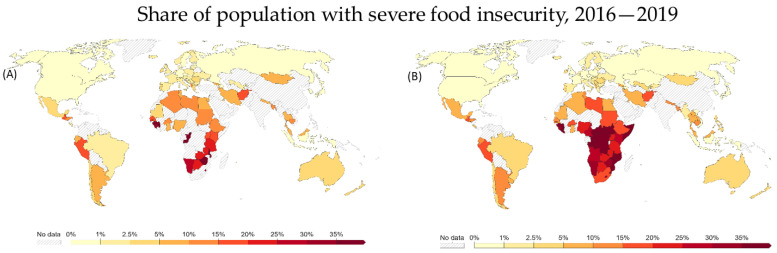
Food insecurity illustrations representing the severity of insecurity from 2016 (**A**) to 2019 (**B**), outlining the increasing nature of food supply issues, particularly in southern Africa where food supply and undernutrition is increasing at an alarming rate [16].

Crop loss as a result of pests and diseases is the primary income loss for farmers and local economies [17]. In a situation where the world can ill afford waste, biotic stress losses compound the impact of a growing population. Crop loss consists of an initial loss through the direct interaction with pests and pathogens, as well as the secondary losses caused by the indirect effects of diseases and nutritional content within the soils, having long-term effects [18]. Therefore, it is difficult to quantify the secondary impacts on crop loss due to a range of random variables that are difficult to account for, such as the prominence of pests and disease at the given time of analysis, in comparison to other seasons, due to the repetitive cycle of growth between harvest and sowing seasons. This gives little opportunity to distinguish the nature of loss between crop generations [19].

Crop yield loss is highly dependent on the treatment of the crops, with fungicides, pesticides, and insecticides being the driving force behind increasing crop productivity. The ability to substitute the use of harmful chemicals with naturally sourced compounds produced by endophytes in-vitro, is expected to have a large impact on the long-term health of all living organisms involved in the process from planting to consumption, whether that be humans or other animals [18].

## 3. Agricultural Potential of Endophytes

Current industrial approaches to combatting stressors are through fertilisers, herbicides, and pesticides which have several questionable effects on consumers and long-term soil quality, where taking a natural approach would offer a less harmful and more consumer-friendly product [20]. Endophytes—both bacteria and fungi—have emerged as highly potent natural resources, which have been utilised to improve stress tolerances of plants post-inoculation and decreased susceptibility to diseases and predators [21].

The overall topic of endophytes is novel, with the main interests in their applications only coming to the forefront in recent decades, these applications being outlined in Figure 3 below. The knowledge of potential solutions that endophytes can provide remains largely incomplete, as defined by the estimated 1 million endophytes on earth and only 100,000 of which have been appropriately described and studied, as of 2015 [1]. Studies conducted in this space typically take an approach of fungal or bacterial analysis, but rarely both. This could potentially undermine or pass over the key complex trends within plants with desirable characteristics. It is important that studies begin to understand the entire “endomicrobiome”—the community of microorganisms that reside within the plant tissue—to understand the complete internal environment and the interplay between species [22]. There are still many unknown elements to endophyte-plant interaction [1], however, a holistic approach provides the potential to give context to the plants internal biome via transcriptomics, proteomics, and metabolomics to understand all aspects of endophyte intervention on plant health [23].

Endophyte presence has shown significant benefits in a number of studies, commonly improving the fresh and dry weights of endophyte-associated plants (EAP) two-fold [24,25,26,27]. As outlined by Singh et al. [28]’s study on the effects of endophytes on zinc and iron deficiencies, EAP consequently gained 2–2.3 times fresh and dry weight as a result of improved nutrient intake abilities. Whereby, the endophyte presence resulted in increased indole-3-aceatic acid (IAA) and phytosiderophore production, which increased nutrient solubilisation significantly [28].

The summary table below (Table 1) outlines the wide variety of reported benefits from the literature reviewed, some of which are known plant pathogens in some species yet provide beneficial impacts to other, unrelated species of plants. The complexities of the genetic factors involved in endophyte intervention strategies must be further researched [29]. It is highly likely that the endophytes listed in Table 1 could have further or different beneficial characteristics when tested on different model plants. This is a key problem that requires solving for large scale approaches to improving crop health.

Recruiting beneficial endophytes in a field application is not as simple as theoretically suggested because the complex relations between the biosystems of soils (rhizospheric microbes and actinomycetes) and plants make for a more difficult introduction of any microorganisms [30]. It has been suggested that the colonisation of a seedling is a convoluted event that is very competitive with other microorganisms and the presence of inhibitors within seedlings [31]. Particularly within the initial 24 h post-sowing where the colonisation by endophytes as early as possible tends to improve a sustained presence in mature plants [32,33]. Inoculation approaches apply to all plant-microbe interactions that cannot undergo vertical transmission, which is relevant for many crop plants such as wheat and grains as they do not readily transmit microbes, posing potent difficulties in the pursuit of large-scale farm-based applications of EAP [33].

There are many strategies currently being utilised to combat micronutrient deficiencies in crops, some of which being the direct supplementation of nutrients, direct genetic intervention on plants, fertilization, and food fortification [34]. Although they each have pros and cons, the endophytic treatment of plants will soon be potentially among the strategies to improve food nutrition. Many of the major diet-based nutrient deficiencies globally (iron, zinc, etc.) have direct connections to the benefits identified from EAP [28,35,36]. Theoretically, endophyte bioprospecting may give rise to new approaches that internally improve food nutrition, without the need for potentially harmful chemical treatments or GMOs. However, significant knowledge gaps must be filled, and the effects of the endophytes must be understood before anything can be deemed safe for human consumption.

**Table 1 microorganisms-11-01276-t001:** Summary table of endophytes reported to have agricultural applications.

Endophyte	Host Plant *	Effect	Reference	Type
*Acremonium* sp. Lp1,2	*Lolium perenne*	Antifungal	[37]	Fungi
*Aspergillus terreus* AM2	*Moringa oleifera*	Antifungal, IAA production	[38,39]	Fungi
*Trichoderma harzianum*	Hardwood bark	Antifungal, P solubilisation	[40,41,42]	Fungi
*Epichloe festucae* Fr1, 11, E365	*Lolium perenne*	Antifungal	[37]	Fungi
*Epichloe* sp.	Not reported	Biotic stress tolerance	[43]	Fungi
*Paraphaeosphaeria sporulosa*	*Actinidia deliciosa*	Biotic stress tolerance	[44,45]	Fungi
*Cochliobolus* sp. 23-1	*Panicum coloratum*, *Chloris gayana*	Ca accumulation	[46]	Fungi
*Ophiosphaerella* sp. 15-2	*Panicum coloratum*, *Chloris gayana*	Ca accumulation	[46]	Fungi
*Penicillium bilaii*	*Elaeis guineensis*	Ca accumulation	[47]	Fungi
*Penicillium oxalicum* P4	*Elaeis guineensis*	Ca accumulation	[47]	Fungi
*Setosphaeria rostrata* GR1A	*Panicum coloratum*, *Chloris gayana*	Ca accumulation	[46]	Fungi
*Fusarium fujikuroi* IMI58289	*Manihot esuclenta*, *Oryza sativa*	GA production	[48,49,50]	Fungi
*Gibberella fujikuroi*	*Manihot esculenta*	GA production	[48,49]	Fungi
*A. lipoferum*	*Zea mays*	GA production	[51]	Fungi
*Penicillium citrinum* IR-3-3	*Ixeris repenes*	GA production	[52]	Fungi
*Penicillium commune* KNU5379	*Seasamum indicum*	GA production	[49]	Fungi
*Penicillium funiculosum*	*Glycine max*	GA production	[53]	Fungi
*A. brasilense*	*Zea mays*	GA production	[51]	Fungi
*Sphaceloma manihoticola* Lu949	*Manihot esuclenta*	GA production	[48,49]	Fungi
*Aspergillus fumigatus* LHL06	*Glycine max*, *Zea mays*	GA production, pathogen resistance	[52,54]	Fungi
*Beauveria bassiana*	*Vitis vinifera*, wheat	General growth, pest resistance	[55,56,57,58]	Fungi
*Colletotrichum tofieldiae* CT04_08450	*Arabidopsis thaliana*	General growth, hormone production	[59]	Fungi
*Diaporthe* sp.	*Festuca rubra*	General growth, IAA production, N, Ca, Mg, Fe accumulation	[60]	Fungi
*Neotyphodium lolii* NEA4	*Lolium perenne*	General growth, Pest deterrence, pathogen resistance excl. staggers	[61]	Fungi
*Acremonium coenophialum*	*Festuca arundinacea*	Growth, pest resistance, pathogen resistance	[62]	Fungi
*Aspergillus flavus*	*Euphorbia geniculata*	IAA production, pathogen resistance	[63]	Fungi
*Piriformospora indica*	Multiple	IAA production, salt tolerance	[64,65]	Fungi
*Gilmaniella* sp. AL12	*Atractylodes lancea*	Jasmonic acid production	[66]	Fungi
*Phomopsis liquidambari*	*Bischofia polycarpa*	N & P accumulation	[67]	Fungi
*Heteroconium chaetospira*	Chinese cabbage	Nitrogen metabolism, general growth	[68]	Fungi
*Aspergillus niger* AP5, P85	*Arachis hypogaea*	P solubilisation, IAA production, Ca accumulation	[47,69]	Fungi
*Glomus mosseae*	Not reported	P, K, Mg, Cu, Zn, Mn accumulation	[70]	Fungi
*Trichoderma arundinaceum*	Not reported	Jasmonic acid production	[71]	Fungi
*Trichoderma harzianum* TRI5	Not reported	Biotic stress tolerance	[72]	Fungi
*Trichoderma virens* IB 119/12	*Glycine max*	Pathogen resistance	[73]	Fungi
*Cladosporium cladosporioides* BOU1	*Solanum melongena*	Pest resistance	[74]	Fungi
*Epichloe coenphiala* AR584	Tall fescue	Pest resistance	[75]	Fungi
*Fusarium oxysporum* 24o, V5W2	*Musa* spp.	Pest resistance	[76,77]	Fungi
*Metarhizium anisopliae* QS155	Not reported	Pest resistance	[41]	Fungi
*Epichloe gansuensis*	*Achnatherum inebrians*	Salicylic acid production	[78]	Fungi
*Bacillus subtillis* 26D	Not reported	Biotic stress tolerance	[79]	Bacteria
*Streptomyces* sp. 11E	*Vigna radiata*	Auxin production, N fixation, salt tolerance, siderophore production	[80,81]	Bacteria
*Azospirillum brasilense* B510	*Oryza sativa*	IAA production, N fixation	[82,83]	Bacteria
*Paenibacillus* sp. ANM59, ANM76	*Cicer arietinum*	IAA production, P solubilisation, salt tolerance	[84]	Bacteria
*Bacillus thuringiensis*	*Zea mays*	Insect deterrent	[85]	Bacteria
*Ewingella americana* EU-M4ARAct	*Zea mays*	K accumulation	[86]	Bacteria
*Pantoea agglomerans* EU-E1RT3-1	*Zea mays*	K accumulation	[86]	Bacteria
*Pseudomonas brenneri* EU-A2SK1	*Zea mays*	K accumulation	[86]	Bacteria
*Mesorhizobium ciceri* BRM5	*Cicer arietinum*	N fixation, IAA production, salt tolerance	[87,88]	Bacteria
*Azospirillum brasilense* Cd, Az39	*Triticum aestivum*	N fixation, P solubilisation, auxin production	[89,90]	Bacteria
*Bacillus* sp. 13E	*Vigna radiata*	N fixation, P solubilisation, auxin production	[80]	Bacteria
*Bacillus endophyticus* 14E	*Vigna unguiculata*, soybean	N fixation, P solubilisation, auxin production, salt tolerance	[80]	Bacteria
*Bacillus altitudinis* Q7	*Ginkgo biloba*	Pathogen resistance	[91]	Bacteria
*Bacillus polymyxa L6*	Not reported	Pathogen resistance	[92]	Bacteria
*Serratia plymuthica* HRO-C48	Not reported	Pest resistance	[93]	Bacteria
*Burkholderia* sp. SSG	*Beta vulgaris*	S metabolism, N fixation, K accumulation, IAA production	[94,95]	Bacteria
*Azotobacter chroococcum* Avi2	Not reported	Salt tolerance, drought stress, general growth	[96]	Bacteria
*Serratia marcescens* AL2-16	*Achyranthes aspera*	Siderophore production, IAA production, ammonia production, general growth	[97,98,99]	Bacteria
*Pseudomonas fluorescens* L228, L111, L321	Not reported	Siderophore production, P solubilisation, general growth, pest resistance	[100]	Bacteria
*Enterobacter* sp. SA187	*Indigofera argentea*	Sulphur metabolism, reduced ROS accumulation, salt tolerance	[101]	Bacteria
*Burkholderia phytofirmans* PsJN	Not reported	Zn accumulation	[102]	Bacteria
*Staphylococcus hominis* 7E	*Vigna radiata*	Zn, P accumulation, hormone production, antifungal	[80]	Bacteria

* “Host plant” refers to the plant from which the organism was been isolated for the first time.

## 4. Plant-Endophyte Relationship

Unpredictable environments consistently place stressors upon plants through abiotic and biotic sources and it is the plants prerogative to efficiently mitigate these stressors to maintain homeostasis [2,3,5,103,104,105,106,107,108,109,110]. The impact that endophytes have on the internal environment and behaviours of a plant is closely tied to the gene regulation and signalling as a result of the endophyte presence [111]. Dependent upon the origin of the stressor, plant-endophyte interactions differ through the type of endophytes present, the signalling mechanism, the secondary metabolites, etc. The effects of the endophytes on plants are constantly being investigated and new depths of the relationships between the two are being uncovered [112]. In the many instances of the significant effects of endophytes on plant behaviours—nutrient uptake improvements, greater stress tolerances, etc.—there are no direct and complete descriptions of the mechanisms by which this is achieved.

Plant-endophyte symbiosis is best demonstrated as the plants observed providing EF with carbon sources, mainly in the form of sucrose as determined by tracing ^13^CO provided to the plant. This evidence supports the symbiotic model between the fungi and plant, where the plant is nurturing the endophytes within them, suggesting that endophyte presence is not of a parasitic nature [68]. Additionally, Usuki et al. [68] identified a negative correlation between *H. chaetospira* and soil nutrition (carbohydrate, nitrogen, and phosphorus), as the plant’s needs are met by the environment the endophyte impact is reduced. This evidence outlines the symbiotic nature of the plant-endophyte interactions, whereby plants rely on alternative pathways of nutrient acquisition when soil nutrition is sub-optimal. This aalso suggests that plants may have a selective permeability to specific endophytes relative to the plants’ internal and external environments [113].

## 5. Role of Endophytes in Plant Health

### 5.1. Biotic Stressors

#### 5.1.1. Pathogens

As aforementioned, primary crop loss is as a result of pathogen and pest intervention impacting agricultural practices to the degree of billions of dollars [114]. Pathogens, in which there are more than 100 differenct types of plant pathogens, can be spread throughout crops via airborne propagation, insect vectors, or via introduced methods such as irrigation [115,116]. Delgado-Baquerizo et al. [117] connected warmer climates to an increased number of pathogens in plants, linking climate change to increased disease levels. An analysis of the entire microbiome can aid in understanding the difference in behaviour between harmful pathogens and the development of beneficial endophytes, as well as explaining any shift in endophyte behaviour, from beneficial to harmful [22].

Endophytes can improve plant biosecurity through the improved antimicrobial activity of EAP, as developed in some commercially available perennial ryegrasses [61]. The majority of literature surrounding endophytic improvement relative to biotic stressors comes from investigating *Epichloe* endophytes, a family of fungi with strong relationships with grasses that have both beneficial and pathogenic effects [78,118]. Beneficial endophytes belonging to *Epichloe* improve biotic resistance through a range of mechanisms, mainly through the upregulation of jasmonic (JA) and salicylic acid (SA) production pathways—as discussed following [78,118,119]. Xia et al. [43] studied the impact of *Epichloe* endophytes on plants infected by powdery mildew (*Blumeria graminis*), suggesting that EF are able to mitigate the chlorophyll loss of activity and damage (measure of disease severity) compared to non-EF containing plants.

Along with antimicrobial properties provided by endophytes, increases in metabolic and photosynthetic rates in EAP help combat the effects of pathogen presence. EAP have shown to have higher rates of growth under pathogen infection, compared to the basal rates of non-EAP [118]. It could be considered that larger and healthier plants have reduced pathogen impact and increased capabilities to mitigate infection [44]. This is exemplified by endophytic increases in calcium (Ca) acquisition, which is involved in the systematic acquired resistance signalling, whereby healthier levels of Ca result in an appropriate ability to respond to stressors [120,121,122].

The antimicrobial properties of endophytes are on display within kiwifruit plants, as their associated endophytes’ can resist canker disease caused by epiphytic bacteria *Pseudomonas syringae* [44,45,123]. *Paraphaeosphaeria sporulosa* is documented as an associated EF to kiwifruit plants (*Actinidia deliciosa*) with antimicrobial activity against *P. syringae* through the production of diketopiperazine [44,124]. Diketopiperazine—a class of organic compounds—are known to have antimicrobial properties, especially those that contain proline, which is a common factor in stress tolerances [125]. Similar results have been reported in wheat pathogens, causing some of the most economically destructive diseases. These diseases, caused by fungi *Magnaporthe oryzae Triticum* (MoT) and *Fusarium graminearum*, can contribute to extraordinary amounts of crop loss [72,79,126]. Chakraborty et al. [79] discovered lipoproteins produced by *Bacillus subtillis* 26D capable of inhibiting MoT growth and germ tube formation in vitro. Taylor et al. [72] outlines that the endophyte *Trichoderma harzianum’s* ability to produce trichodiene—a known biocontrol agent—is a sesquiterpene that inhibits *Fusarium graminearum*, the pathogen responsible for fusarium head blight.

The complexities of endophytic communities can be demonstrated through the disease ryegrass staggers, a common condition in sheep, cattle, and horses during winter as grazing approaches roots containing *Neotyphodium lolii* [127,128]. Alkaloid toxins produced by *N. lolii* are harmful to livestock but can be beneficial to the plant resistance to stem weevils through the same mode of action [127,129]. It is the complex balance between benefit and cost that must be assessed to responsibly apply these novel techniques. Increasing the biomass of crops reduces the amount of *N. lolii* containing roots consumed by livestock, but does not mitigate it entirely, therefore it is currently a case-by-case scenario for applying these techniques.

Examples, such as those aforementioned, provide validity to the theoretical applications of endophytes as biocontrol agents. However, there must be more research surrounding the impacts of the introduced species on the natural biome of a plant, alongside the cost to the plants and any downstream consumers in order to ensure safety of the practice [130,131]. As there are many secondary metabolites produced by the endophytes, there is always potential that some of these compounds can negatively impact the plants and consumers if they are left to bioaccumulate [131].

#### 5.1.2. Secondary Metabolites

The metabolic profiling of endophytes in recent years has identified a wealth of secondary metabolites with antimicrobial, anticancer, and pesticidal properties. Including compounds such as flavonoids, carotenoids, melatonins, terpenoids, phenolics, alkaloids, peptides [132,133]. 42% of the newly approved drugs between 1981–2019 are of, or a derivative of, natural sources, many of which are produced by endophytes [134]. Although the large majority of the metabolite discoveries are found in the context of medicinal applications, the same bioprospecting principles can be used in the context of agricultural practices [133]. Many metabolites with agricultural applications are multifunctional in terms of plant growth promotion (PGP), along with improved abiotic or biotic stress tolerances.

Endophyte-derived bioactive compounds such as acyl-homoserine lactones, diacetylphloroglucinol (DAPG), and pyoluteorin, have been utilised in managing soil-borne pathogens, suppressing weeds, and PGP [135,136]. The application of these compounds to crops offers the potential for the development of eco-friendly, low-cost, and sustainable agricultural practices. Similarly, non-ribosomal peptides and polyketides have been identified as components of the systemic acquired resistance (SAR) in plants, which can strengthen the host plant immunity to various pathogens [137]. Additionally, amongst these secondary metabolites are plant hormones, which stimulate the germination of seeds, root growth, and nutrient uptake [101]. The origin of many of these metabolites are not well understood, as evidence outlines relationships where the compounds are produced by plants, microbes, or co-production between both organisms. Therefore, an understanding of the endophytic secondary metabolites and the pathways by which they are produced is vital to enhance plant productivity and resistance to pests in agricultural practices.

#### 5.1.3. Pests

Complementing pathogen resistance, endophytes can impact the other half of biotic stress tolerances, which is pest resistance [138,139]. Volatile organic compounds (VOC) produced by plants can have a variety of pesticidal functions, they can be directly toxic, a repellent, or an attractant for pest pathogens (caterpillars, spiders, etc.). The term VOC is a broad umbrella term encompassing compounds such as pheromones, phenolics, and alkaloids [139,140]. It is widely reported that endophytes produce VOCs as well as stimulating VOC production within plants via the jasmonic pathway (JA). Evidence shows that EF stimulating this pathway through the production of key compounds in the signalling cascade, thus increasing resistance to pests through antiherbivore VOC production [141,142]. Endophytes that possess this ability are a viable biocontrol agent in principle. Unlike many other stressors, biotic resistance is one of the only stress tolerances that currently have field applications in the form of EAP [143,144]. Commercially available ryegrass strains AR1, AR37, NEA2, and NEA4 are examples of EAP to improve yield, increasing lateral root growth by up to 92%, and significantly increasing the number of tillers aiding in persistence. Furthermore, these strains show increased resistance to weevils and black beetles by approximately 75% and 60%, respectively [61].

The most studied and widely available biopesticide is the use of *Bacillus thuringiensis*, a pore-forming toxic protein producing EB with insecticidal characteristics. Cry toxin expression in crops has shown to improve pest resistance, as it impacts the larval stages of many phyla of insects [85]. The pore-forming toxins target the digestive tracts of the larvae through midgut cell membrane insertion resulting in osmotic shock. *B. thuringiensis* products are commercially available as sprayable products and take up 35% of the biocontrol agent markets [77]. Second to the aforementioned, *Beauveria bassiana* is the next most studied endophytic biopesticide, and is an EF capable of parasitising more than 200 insect species [57]. Wang et al. [145] describes in great detail the production of degrading enzymes by which *B. bassiana* is able to infiltrate pest hosts, followed by the production of the insecticidal compounds beauvericin, bassianin, bassianolide, beauverolides, tenellin, oosporein, and oxalic acid, all of which contribute to the interference with cell functions and antimicrobial responses within the host insect [146,147]. Similar insecticidal characteristics have also been reported from other entomopathogenic microbes, including *Rhizoctonia solani*, *Fusarium* sp. [77], *Epichloe coenophiala* [75], *Pseudomonas fluorescens* [98,99], *Serratia plymuthica* [93], *Cladosporium cladosporioides* [74], and *Metarhizium anisopliae* [41].

The discussed endophytes represent a proof of concept of endophyte-based approaches to be implemented into agricultural practices. Not only as biocontrol agents, but to open the door for human-consumed crops to receive treatments relative to nutrient deficiencies and abiotic stress tolerances to improve yields.

### 5.2. Phytohormones

#### 5.2.1. Auxins (Indole-3-Acetic Acid)

Endophyte presence often promotes plant growth as a result of the phytohormone production or upregulation, and is the most widely known mechanism for physiological and structural changes [110]. The production of phytohormones is a direct relationship to growth signalling. Hormones produced by plants, EB, and EF—such as Indole-3-acetic acid (IAA)—can stimulate taproot/adventitious root growth [148]. IAA is a multifunctional signalling molecule that transcriptionally effects small, nuclear, short-lived proteins resulting in a range of auxin-regulated transcription throughout the cells [149]. The introduction of phytohormone-producing EB and EF to plants in stressed conditions has shown clear persistence across a range of stress parameters [150].

*Piriformospora indica* is a key EF identified to produce IAA as a secondary metabolite, and is one of the most commonly studied EF as a plant inoculant [65,151]. Auxin signalling has been reported to be involved in iron solubilisation by inducing Fe (III) reductase in *Malus xiaojinensis*. Furthermore, IAA signalling is associated to 1-aminocyclopropane-1-carboxylate (ACC) deaminase activity, an important enzyme involved in limiting the effects of salt stress [152,153]. ACC deaminase degrades ACC, a precursor to ethylene, which is a plant hormone responsible for the reactive oxygen species (ROS) accumulation under salinity stress [153,154]. Of the literature reviewed and current knowledge base, endophytes capable of producing auxins have a strong relationship with improved plant growth, one of the strongest impacts on overall plant behaviour and physiology [110,155].

#### 5.2.2. Gibberellins

Gibberellins (GA), like auxins are a class of growth promoting phytohormones involved in a vast array of effects on plant cells, particularly stem elongation, flowering, and leaf spread. 136 GAs have been discovered, only a few of which meet the criteria of bioactivity (containing 19 carbon backbone and hydroxyl group), GA_1_, GA_3_, GA_4_, and GA_7_ have been identified as the most bioactive of the GAs [48,156,157,158], however, GAs that are considered ‘inactive’ are precursors to bioactive GAs with subtle differences, which can be assimilated enzymatically to form bioactive GAs [150,157]. GAs are regulated by DELLA proteins via a negative feedback loop. DELLA repression due to the binding of DELLA and GID1—a GA receptor—blocks signalling until the repressor is ubiquinated by ubiquitin-transferase SCF [159]. There is evidence to suggest that endophytes play a role in the transcription of DELLA and SCF proteins, in turn improving GA effects alongside any GAs produced as secondary metabolites [101]. GA-producing endophytes have been characterised down to the presence or absence of GA gene clusters. Endophytes that possess these genes typically belong to *Ascomycetes* phyla [150]. Association with GA producing endophytes has shown to benefit plants, yet the direct mechanisms are yet to be appropriately described [150]. Identifying a key trend in the characteristics of a taxonomical classification—contrary to the majority of endophyte knowledge—are rare in the current climate of understanding characteristic trends relative to the beneficial properties that endophytes can bestow on plants.

#### 5.2.3. Stress Response Hormones

Three phytohormones commonly associated with a range of stress responses can be collectively termed as the stress response hormones, and include abscisic acid (ABA), salicylic acid (SA), and JA [160]. Current knowledge of these three systems suggests a complex interplay between the three signals and that the presence of endophytes only entangles the relationships more [161,162].

##### Abscisic Acid

Phytohormone ABA is commonly known as the plant stress hormone, involved in major signalling pathways involved in biotic stress response, as well as being responsible for general plant growth outside of the stressed conditions [163]. Under drought stress, ABA is produced rapidly by plants to aid in the plant’s survival in sub-optimal conditions. Increased levels of ABA stimulate stomatal closure through H_2_O_2_ production in order to preserve moisture, as well as decreasing lateral root growth and increased vertical (main) root elongation to vitally improve water uptake, and accelerate sentience [163]. However, contrary to early theories, endophyte association negatively correlates with ABA levels, as shown by Waqas et al. [164] who reported a negative correlation of ABA and *Penicillium* sp. and *Pterolepis glomerata*. Khan et al. [150] suggest that this indicates that ABA is an independent mechanism of stress mitigation. Conversely, Wang et al. [161] reported positive correlations between EB and ABA levels, suggesting that conjecture remains surrounding the effects of endophyte presence on ABA levels, posing the potential for ABA-dependent and ABA-independent pathways. Despite these findings, it must be considered that the negative relationships have been more heavily reported. This is potentially due to the salinity and drought stress that the plant finds itself under; ABA produces H_2_O_2_—a ROS—which is undesirable under saline conditions, and therefore reducing levels of ABA is a likely plant resistance mechanism to minimize ROS accumulation [163].

##### Salicylic Acid

Although endophyte presence typically reduces the ABA levels in plants, theoretically reducing the drought tolerance levels of plants, the SA system can counter for the reduced defence mechanisms that ABA provides [6]. SA, a phytohormone capable of initiating the systematic acquired resistance (SAR), provides a whole plant response to the localised infection by pathogens [165]. The SAR is a wide affecting system much like the innate immune system in animals. Plant SA production occurs through isochorismate synthase and phenylalanine ammonia-lyase pathways, however, not all steps of these pathways have been discovered [78,166]. Incomplete knowledge of these mechanisms creates a gap when assessing the impacts of endophytes on SA production. Understanding the plant mechanism in its entirety can potentially uncover key compounds and enzymes involved in the pathogen resistance in EAP [167].

Kou et al. [78] assessed the effects of *Epichloe gansuensis* on *Achnatherum inebrians*, its ability to increase the production of SA, and its pathogen resistance relative to endophyte presence. EAP showed increased levels of SA and the transcription of genes relevant to the SA pathway. Increased SA production induces a pathenogenic-related gene transcription, involving bacterial and fungal lysing enzymes [168]. The mode and degree of action of the SAR is relative to the specific signalling and context of each cell, making it difficult to accurately quantify the impact of the variables, such as endophytes and other phytohormones [169]. Acibenzolar-S-methyl is the most studied analog of SA and is a key inducer of the SAR, however, its function remains incompletely understood like most of the SAR. The literature is limited to a broad description of the SAR and remains vague when explaining the mechanisms by which plants act against pathogens. Evidence suggests that the SAR induces pathogenesis-related proteins [168,169]. This is due to the large number of proteins and genes involved in the pathogenesis and the SAR, which include antimicrobial peptides, other phytohormones, protein & MAP kinases and transcription factors [170].

##### Jasmonic Acid

Jasmonic Acid (JA) is a key phytohormone involved in a wide variety of cellular regulation from senescence, flowering, and abscission. JA is also the key signalling molecule involved in stress signalling as a result of plant wounding from pests. More specifically, JA is released when the plant is consumed, stimulating *CsKPI* genes responsible for Kunitz protease inhibitor (KPI) expression—a known anti-herbivore compound [141,142,171,172]. Protease inhibitors, when consumed, prevent the digestive enzymatic activity of insects which reduces the metabolic potential of the insect, thus, making the plant undesirable as a source of food and increasing pest (insect) resistance as a result of the increased levels of JA [173]. *CsKPI* genes have been the subject of genetic engineering, however, the conjecture and limitations surrounding GMOs leaves an opening for alternative methods to promoting genetic transcription, including endophytes [171].

Endophytes have shown clear implications on the JA levels of plants, often antagonistically to SA pathways [173]. The direct treatment of plants with JA are sub-optimal due to this relationship with SA, as it leaves plants susceptible to viral and bacterial infections [174]. Endophyte *Gilmaniella* sp. provided evidence that endophytes produce JA and VOC in *Atractylodes lancea* [66]. Mengistu [71] explains the pathway by which EF *Trichoderma arundinaceum* stimulates the JA related defence genes, achieved through the production of trichodiene [72,175]. Trichodiene, the forementioned compound, with activity against *F. graminearum*, which is the pathogen responsible for wheat blight [72].

The complexities of the interrelationships of the ‘stress signalling hormones’ (JA, SA, ABA) have only recently begun to be appropriately understood. Yet the complete pathways of each hormone—both irrespective and respective of each other—must continue to be described [161,162]. This is in order to most appropriately apply the theories of endophytic inoculation on agricultural practices.

##### Cytokinin’s

Cytokinins (CK) are a class of phytohormones involved in apical dominance and lateral root growth signalling which directly act as an antagonist to auxins. Recently, CKs have been the subject of studies regarding pathogen resistance, with evidence suggesting that CK are involved in regulating the SA pathway, as well as SA independent resistance [167,176,177,178]. Großkinsky et al. [178] identified that the endophyte G20-18 was capable of controlling the forementioned *P. sryingae* infection, when CK levels was adequate. Reduction in these levels diminished the capabilities of G20-18 to control the bacterial pathogen [178]. It has been reported that parthenogenesis occurs by regulating the host CK production, and external sources of CK such as endophytes offer alternative routes to pathogen suppression via CK [179,180]. The interference of pathogen development as a result of endophyte presence has been identified to occur amongst the cytoskeleton and cell trafficking of pathogens. Gupta et al. [181] tested CK and derivatives directly against *B. cinerea* and found that zeatin, kinetin, thidiazuron, and 6-benzylaminopurine were able to inhibit the growth of the fungi. The suggested mode of action is the interference of proliferation through spores and tube elongation.

The well-known aspect to CK is also relevant to benefiting from endophytic association, as higher levels of CK results in the stimulation of lateral root growth and an overall increase in biomass [151,167]. This is the case with *P. indica* association, whereby plant biomass is significantly increased due to its presence as a result of higher CK levels [167]. As aforementioned, endophytes have the capacity to produce phytohormones as well as stimulate the transcription of them. This is the case with CK where some endophytes have the capacity to produce their own CK to improve plant growth [164,182].

##### Reactive Oxygen Species

ROS are derivations of oxygen that are more reactive than molecular oxygen (O_2_), and are signalling molecules that are involved in plant senescence, degrading cellular components which are no longer required, and maintaining redox homeostasis. Under stress conditions, ROS production increases significantly and can surpass the metabolism capacity of osmolytes which results in accumulation [183,184,185]. Accumulation of ROS caused by stress can seriously damage the cellular components such as DNA, RNA, and proteins, potentially resulting in cell death [186]. H_2_O_2_, super-oxide (O_2_^−^), hydroxyl radicals (HO^−^), and nitric oxide (NO^−^) are all ROS, where plants combat the accumulation through antioxidants such as proline, peroxidases (POD), superoxide dismutase (SOD), catalase (CAT), gamma-aminobutyric acid (GABA) [183].

Endophyte presence has shown to positively impact the osmolyte and antioxidant production within plant tissues, providing an improved ability to metabolise ROS [119,187]. Studies investigating the inoculation of *B. subtillus* and its effect on stress tolerance improvements found that the EB stimulates CAT, SOD, and POD production in chickpeas. As a result, plants experience greater tolerance under salt stressed conditions, however, these results were observed in both saline and optimal conditions and is therefore not specific to stressed conditions in these circumstances [188,189]. Contrary to this, a study of glutathione peroxidases (GPX) conducted by Bela et al. [185] identified genes relevant to GPX production activated by oxidative stress, that resulted in reduced cell death caused by heat and salinity, and could potentially act as cellular redox sensors.

As Khan et al. [189] identified, endophyte presence can increase antioxidant production in all conditions, increasing the basal level of antioxidants. Compounded with oxidative stress-induced reactions to salinity which further increases the overall defence to ROS. Improved nitrogen availability has shown to improve salt tolerance due to increased proline biosynthesis, as proline is a secondary amine [190]. The ability of the endophytes to enhance bioavailable nitrogen to plants provides another direct link from endophyte-based improvements and increased stress tolerance.

Hormones produced by plants and microbes are involved in a complex model of regulation and identifying the hormone levels in plants is one of the best insights into the context of a plant’s health. Table 2 below summarises the key effects of each plant hormone affected by endophyte presence.

### 5.3. Nutrient Limitations

#### 5.3.1. Iron

One of the most common deficiencies in plants is Fe deficiency. Fe is involved in the production of chlorophyll, is a common enzymatic co-factor, and plays a major role in the electron transport chain (cellular respiration) [197]. Alkaline soils make up approximately 30% of the farmable land and cause Fe deficiencies more than any other condition, therefore overcoming this limitation greatly improves the ability to farm on marginal land [198]. Graminaceous plants—most grass and cereal species—respond to conditions of Fe deficiency by producing phytosiderophores (PS), a class of organic substances that can solubilise rhizospheric Fe and Zinc (Zn), known as strategy I [6,35,199]. Some endophytes produce and excrete PS around the roots of non-graminaceous plants, solubilizing Fe (III) through the binding of mugineic acids (MA), conjugating to form Fe (III)-MA which resolves the requirement of reducing Fe (III) to Fe (II) for plants to absorb rhizospheric metals as outlined in Figure 4 (strategy II) [36]. Studies have suggested that this mechanism of PS-involved metal transport can solubilise other metals such as Zn, copper (Cu), cadmium (Cd), and Ni [200]. Thus, identifying a direct link between endophyte presence and an improved response to metal deficiencies via PS production. Fe plays an important role in a number of cellular processes downstream and Fe deficiencies extend beyond direct Fe cellular processes, impacting plant growth across the board [28,201].

The direct mechanisms of impact of endophytes on Fe accumulation are poorly understood. Tt has been speculated that Fe deficiency stimulates ROS accumulation, resulting in a suppression of Fe-related and regulated genes *FIT*, *IRT1*, *FRO2* [28,202,203]. Additionally, endophytes have been observed to stimulate IAA induced Fe(III) reductase activity, aiding in Fe accumulation via strategy I [192]. Thirdly, endophytes have been observed to increase PS levels in plants, giving strategy II capabilities to [109,201]. Regardless of the mode of action, endophytes have shown to play key roles in improving Fe availability and represent a potential solution to the most common consumer nutrient deficiency [28,46]. However, reported results originate from the investigations of relationships between individual endophytes and plants. To appropriately analyse and describe the effects of endophytes on Fe acquisition, a larger knowledge base is required to assess trends across phylogenetic groups of both plants and EF and EB, for there is potential for multiple points of injunction on plant processes.

#### 5.3.2. Zinc

Zn behaves similarly to that of Fe relative to plant nutrition; typically, deficiencies occur in calcareous and clay containing soils due to adsorption of reactive Zn^2+^ ions. Zn is the second most organically abundant transition metal after Fe [204]. Deficiencies result in chlorosis, stunted plant growth, and reduced cellular functions as Zn is an important enzymatic co-factor [204,205]. It is important to ensure the mitigation of dietary inadequacies in consumers down-stream, and nutritionally fortifying the crops that are most susceptible to Zn deficiency, such as corn, provides a solution to undernutrition.

Zn^2+^ can be acquired by graminaceous plants through two strategies, the first of which is the chelation pathway, which also falls under strategy II nutrient acquisition [6]. Chelation is the process by which plant or endophyte exudates (typically polydentate, or multi-dentate, ligands) bind with the charged ions (such as Fe^3+^, Zn^2+^, Ca^+^, and magnesium (Mg^2+^)) to form chelate complexes. Chelating agents are amino acids (glutamic acid & glycine) and organic acids (Ethylenediaminetetraacetic acid, Peracetic acid, Diethylene Triamine Pentaacetic Acid, Nitrilotriacetic acid, Hydroxyethylethylenediaminetriacetic acid, & Ethylenediamine) that scavenge metal ions from the soil [206,207]. Although not the primary function, the PS—MA pathway is capable of taking up other cations, although its affinity remains firmly targeted towards Fe [36,200]. Secondly, plants can absorb Zn in ionic form (Zn^2+^—stable due to a complete outer shell) with the use of ion-transporter channels at the expense of ATP (active transport) [205]. As aforementioned, chelating compounds can be produced and excreted by some endophytes, and these exudates can aid in accumulation via the lower energy pathway of complex formation [36]. There is, however, the potential for hyperaccumulation as a result of endophyte presence as a possibility and excess zinc is toxic to plants. Where Zn accumulation occurs outside of cellular regulation (EAP), plants can be unable to ‘turn off’ accumulation, and endophyte excreted PS could continue to accumulate Zn resulting in the toxification of the plant. It is unknown whether appropriate regulators are involved in the PS accumulation pathway [204,208].

*P. indica* has proven to show improvements in Zn accumulation in wheat grown under Zn deficient conditions, increasing overall vegetative weight, and branching of the roots. An increase in POD levels was observed in EAP and under oxidative stress it can be theorised that EAP would tolerate ROS better than non-EAP as a result of adequate Zn [183,209]. Highlighting the overall complexity of plant nutrition and the interwoven nature of plant systems, tying many metabolic functions together due to common factors such as Zn [209]. The aforementioned literature suggests that endophytes can significantly improve Zn uptake leading to improved plant growth.

#### 5.3.3. Nitrogen

Nitrogen (N) is a key nutrient to plants as it is a major component of chlorophyll and amino acids, and in turn impacts the energy production and protein functions of cells [196]. The presence of endophytes has shown to significantly alter the chemical processes of basic plant survival mechanisms, outlined by a 2007 study that analyses the propensity of *Heteroconium chaetospira* to provide Chinese cabbage plants with nitrogen that plants are typically unable to access [68]. *Pinus contorta* has been observed to thrive in extremely low N conditions, alerting Padda et al. [210] to a potential alternate nitrogen source, with the likely cause being symbionts. 14 endophytes in this study were assessed as inoculants of *P. contorta* and were found to significantly increase seedling size and biomass under N-limited conditions. Most importantly, 11 endophytes increased free nitrogen in the plant by >200% and it was concluded that EB are able to fix N from the atmosphere to help *P. contorta* thrive in N-limited soil. High levels of nitrogenase were reported in endophytes isolated from *P. contorta*, an enzyme that classifies EB as nitrogen-fixing and explains the mode of action of these results [210,211]. There are many studies that have identified endophyte association improving N bioavailability through N^13^ radioisotope tracing from atmospheric N [68].

The significance of N to overall plant health cannot be understated as it plays a role in almost all cell functions through proteins [196]. Improving N availability through endophyte presence or fertilisation has a large flow-on affect to overall plant health. One example demonstrated in Figure 5 is how N availability can apply to drought resistance by metabolising ROS with proline, and in turn reducing accumulation and allowing IAA synthesis to continue uninhibited. [163,183,190,196]. Table 3 (page 19) summaries the impacts of plant nutrition on overall plant health as these tie closely to the theory of the vast downstream effects outlined in Figure 5.

N fertilizers, such as urea, are common practice to improve crop yields, as they are the most effective crop health improvers. However, urea can be harmful to the environment and livestock if inappropriately treated [212]. Using alternate sources to improve N bioavailability, such as EAP, greatly reduces the threat to the environment and the effort required to sustain healthy crops [213,214].

#### 5.3.4. Phosphorus

Crop production is commonly limited by phosphorus (P) bioavailability due to phosphate formation, and the complex fixations with Fe, Ca, and aluminium (Al) [67]. A study conducted by Varga et al. [215] concluded that endophyte association (WP5 and WP42) can increase P solubilisation from Ca_3_(PO_4_)_2_ and AlPO_4_ in *Poplus* trees 80 and three-fold, respectively. The proteomics of plants with versus without sufficient P showed a general increase in metabolic related proteins, particularly those involved in sugar metabolism, of which P is an important co-factor. The increased metabolism rates are likely responsible for the increased dry weight of plants observed in the study. Multitudes of studies similar to Varga et al. [215] are required to understand the overall trends throughout plant endophyte relations, taking ‘omics’ approaches aid in providing context to the physiological impacts intracellularly.

Unlike many other essential plant nutrients, the role of P in plants—particularly agricultural plants—has been heavily studied as it is one of the greatest limiting factors in crop production, as fertilizers like the aforementioned urea are currently used to improve P bioavailability [67,213]. P fertilisers are of low efficiency due to P fixation and pose a great risk to the environment as they can contain toxic elements such as lead (Pb), mercury (Hg), and arsenic (As) [216,217]. Endophytes possess the potential to remove harmful contaminants from bioaccumulating in crops or livestock through reducing current fertilization practices. The large amount of agricultural land is saturated with P from previous fertilization, but remains fixed to Fe, Ca, and Al, as EAP inoculated with the right strain can access this soil-bound P and reduce the need for fertilization [67].

Endophyte association increases P solubilisation by increased gluconic, malic, citric, salicylic, and benzene acetic acid production and phosphatase activity [218,219]. Kumar et al. [220] investigated the impacts of *P. indica* presence under P limited conditions which resulted in a 150% and 20% biomass increase relative to P-limited and non-P-limited conditions, respectively. Similarly, studies indicate that endophytes can mitigate P-limiting conditions, with both control and P-limited plants resulting in a similar biomass and growth parameters [221,222]. Suggested effects of EAP are an increased production of organic acids in root associated microorganisms, as acidification of root environments aid the solubilisation of fixed P in soil [218,223]. Endophytes provide a strong alternative to crop acquisition of P which is typically delivered in a particularly problematic method.

#### 5.3.5. Potassium

Potassium (K) is one of the most common elements in the earth’s crust, however, bioavailable K in soil is relatively low. The large majority (90–98%) of soil K is unavailable due to fixation, as K plays a key role in soil structure [224]. K^+^ is a vital cation involved in a number of cellular processes, including but not limited to metabolic regulation, photosynthesis, stress signalling, and maintaining redox homeostasis [224]. Extracellular excretion of MA involves symport membrane proteins involving K^+^ in the exchange [225,226], which ties K deficiency to an inability to obtain Fe via strategy II (the lower energy required pathway). K is also a key co-factor in carbohydrate metabolism, as it is an enzyme activator of carbohydrate metabolising enzymes impacting the general mass, growth, and reproduction of the plant [227,228].

Due to limited amounts of bioavailable K, attaining the maximum potential from the soil is crucial in maximising yields. Endophyte *Burkholderia* sp. significantly increased K availability from a range of fixed sources. To the degree that additional K supplementation did not have a statistically significant impact on the bioavailable K, as endophytes were able to provide sufficient levels of K [219]. However, it is possible that EAP cannot make up for soil K shortcomings and may need to be used in conjunction with current fertilisation practices to improve crop management procedures. Even if endophytes reduce the net requirements for fertilisation, it can be seen as a stabilisation of agricultural land, removing a proportion of the threat from the environment. Isolated endophyte studies have also identified the ability of endophytes to solubilise K in media, clearly indicating the quantitative measure to which endophytes can improve nutrient solubilisation. Rana et al. [86] demonstrates that endophytes identified as K solubilising—*Pseudomonas brenneri*, *Ewingella americana*, and *Pantoea agglomerans*—showed to improve root length, Fe concentration, protein, and chlorophyll content in-planta. This provides solutions to a number of deficiencies and plant short comings, proving the importance of adequate K in plants.

#### 5.3.6. Sulphur

Sulphur availability is also an important factor in plant growth, as it is involved in the production of proteins that can impact the physiology and structure of plants via important amino acids cysteine and methionine (Figure 6) [25,94,229]. Sufficient levels of cysteine and methionine provide sufficient building blocks for polyamines and POD which are the antioxidants involved in the detoxification of ROS [101]. Identifying a link between endophyte impact on the transcriptome and tolerance to salinity stress is a key factor in agronomical crop failures [186].

A 2021 study of EB in sulphur limited conditions observed a significant abundance shift in bacteria of the *Burkholderia* genus under sulphur limited conditions [94]. A more in-depth study of *Enterobacter* sp. SA187 in *Arabidopsis* identified the mechanisms by which the particular EB interacts with the host plant to improve sulphate tolerance [101]. The study concluded that the treatment of sulphur-deprived plants with SA187 showed improvements in sulphur metabolism by the same degree as treatment directly with sulphur supplementation, by impacting the ethylene signalling pathway.

Improved sulphur metabolism as a result of endophyte presence is due to transcriptional regulation impacts of EB, which expand potential environmental sulphur sources to include sulphate and alkane sulfonates [94]. Transcriptome analysis of *Arabidopsis* identified 873 up-regulated and 610 down-regulated genes as a result of SA187 presence, outlining the significant and intricate impacts that endophytes have on plants, due to the volume of genes impacted, it is likely that SA187 benefits plants in other micronutrient deficiencies [101]. Although there is significant evidence to suggest that these transcriptome changes are directly tied to the benefits, radiolabelled tracing of sulphur would help quantify the direct physiological impact on the plant.

#### 5.3.7. Calcium

*P. indica* shows great potential to improve multiple nutrient acquisitions in plants, one of which is Ca [122]. The determinant factor of cell wall integrity and rigidity, and key growth regulator, Ca is important for the fundamental structures of healthy plants. Ca^2+^ is also involved in secondary messaging, including a broad spectrum of functions, such as biotic and abiotic stress responses. Being one of the first signals in the pathogen defence cascade of the SAR [120,121,122]. *P. indica* [122], *Penicillium bilaii*, *Penicillium oxalicum*, *Aspergillus niger* [230], *Ophiosphaerella* sp., *Cochliobolus* sp., and *Setosphaeria rostrata* [46] all show the capacity to improve Ca solubilisation in-planta. Ca acquisition correlates heavily with P and Mg solubilisation due to the common source of Ca originating from calcium phosphate and dolomite [46,231]. Ca is taken into roots through permeable ion channels via active transport, and endophytes improve intake by stimulating this process. Endophytes also improve nutrient uptake by increasing the capacity of accumulation through root growth stimulation [232]. The improved solubilisation of Ca by endophytes is not completely understood and requires further research collectively with P solubilising EB and EF to understand the full picture [230].

#### 5.3.8. Magnesium

Mg is involved in growth regulation, cell wall structure, chlorophyl concentration, and photosynthesis rates as an enzymatic co-factor [26,233]. Furthering this point, evidence suggests sufficient Mg levels are correlated to a number of active sites of enzymes responsible for the acquisition and transport of other nutrients [234]. Mg helps biotic tolerances by improving the structure of cell walls, moreover, Mg^2+^ has been described to act competitively for binding sites of pathogens [235,236]. Similar to other cations such as Fe, Zn, and Ca, Mg is taken into plants by one of two pathways, active transport, or chelation (strategy II) [36,206].

Endophytes *P. indica* and *Lolium perenne* show improved Mg acquisition flowing on to impacting general growth and activity of plants [103,233,237]. Prasad et al. [237] details that *P. indica* has a positive relationship with *Arabidopsis thaliana* chlorophyll concentration both under Mg deficient and sufficient conditions. This uggests that endophytes are capable of overcoming nutrient deficiencies, whcih is due to the aforementioned increases in chelating compounds produced by endophytes [36,206]. Moreover, utilisation of Mg is higher in EAP resulting in higher levels in the plant tissues [103,206].

Theories and evidence suggest that endophytic improvement of Mg, and all nutrients aforementioned, presents a flexible application of EAP. Whereby endophytes can provide stress tolerances in conditions of low nutrition, as well as providing improved nutrient acquisition, transportation, and utilisation under optimal conditions, improving gross yield and biomass [233,237].

**Table 3 microorganisms-11-01276-t003:** Summary table of effects of endophytes on key plant nutrients.

Nutrient	Mechanism of Endophyte Intervention	Effect on Plant	Reference
Iron	Phytosiderophore production—reducing the energy requirements to transport Fe into roots.	Reduces ROS accumulation. Key roles in photosynthesis and the electron transport chain.	[28,36,197,202,203,218]
Zinc	Phytosiderophore production—as above.	Enzymatic co-factor in root growth. Component of chlorophyll.	[28,36,204,205,209]
Nitrogen	Atmospheric fixation when soil N is low (mainly bacteria due to fungi lacking nitrogenase).	Key factor in amino compounds. Increases proline concentration.	[67,71,82,95,186,210,238]
Phosphorus	Organic acid production & soil acidification to increase solubilisation, and increased proteins involved in sugar metabolism.	Root growth stimulation. Key component in DNA and RNA.	[67,215,216,218,219,222]
Potassium	Improving flow of free K^+^ into roots. However, remains limited by available soil K.	Metabolic regulation, photosynthesis, stress signalling, and maintaining redox homeostasis.	[25,219,224,225]
Sulphur	Protein production to utilise sulphate and alkane sulfonates.	Detoxification of ROS. Key structural component of proteins.	[94,101,229]
Calcium	Tied to P and Mg solubilisation due to complex formation	Maintains redox homeostasis. Key structural component of cell walls.	[53,120,148,230,231]
Magnesium	Not determined	Key co-factor to enzymes involved in a diverse range of metabolic functions.	[234,235,237]

## 6. Large-Scale Agricultural Application

The continuation of endophytic studies is required to analyse the stability of inoculated plants in a large scale/commercial setting beyond current applications. Endophytes have been successfully incorporated into commercialised perennial rye grasses on several occasions, improving the performances of grass growth on an annual scale [61]. The knowledge of currently available endophyte-treated products is protected behind patents and intellectual property rights, making it difficult to critically analyse the effects of endophyte presence [111]. Viable results stemming from EAP in grasses is highly relevant to providing validity to proposing endophyte-based applications to agroeconomical crops, maximising yield whilst maintaining surveillance of adverse effects on the livestock consuming such products [239]. At the time of writing, there is no reported negative side effects from EAP consumption within animal trials, yet the costs of endophyte inoculation to plants and humans are yet to be studied and described appropriately. It is vitally important that all aspects of EAP are understood to consider them as safe treatments for consumption by the general public.

Applying the theory of commercial rye grasses to hypothesise the successful incorporation of endophytes into human consumed crops, many aspects of food supply can be improved. Endophytes that are capable of increasing biomass through a range of mechanisms aid in the total yield of produce from crops, improving supply security. Furthered by abiotic stress tolerance increases allowing for greater yields in sub-optimal conditions and opening new land for agricultural use. Additionally, biosecurity is increased through increased biotic stress tolerances in EAP, reducing crop loss to pests and pathogens. EAP can also be an important strategy to combat malnourishment globally, improving nutrient levels within widely consumed crops (rice, wheat, cereals) as a form of dietary supplementation. Globally, 742 million tonnes of rice are produced per year, a marginal increase in yield efficiency can result in an extraordinary change in our abilities to provide for malnourished populations. Endophytes possess the ability to increase gross yield through increasing efficiency per hectare and increasing total farmable land.

## 7. Conclusions

In conclusion, this review paper has highlighted the significant role that endophytes can play in promoting plant growth and improving nutrient acquisition in agriculture. The interaction between endophytes and their host plants has been shown to result in enhanced nutrient uptake, stress tolerance, and disease resistance, ultimately leading to improved crop yields. Furthermore, endophytes offer a promising and eco-friendly alternative to traditional agricultural practices, reducing the need for synthetic fertilizers and pesticides. Research in this area is ongoing, with numerous endophytes being identified and characterised every year in an attempt to develop practical applications for their use in agriculture. As such, the potential of endophytes as a sustainable solution for improving crop productivity and reducing environmental impacts cannot be understated.

Despite the significant progress made in understanding the role of endophytes in agriculture, there is still much to be discovered. Continued research is needed to gain a deeper understanding of the interactions between endophytes and plants, as well as the factors that affect these interactions. This will involve identifying and characterizing new endophytes, studying their modes of action, and understanding their effects on plant growth, and nutrient uptake under different environmental conditions. Additional research is needed to develop practical applications for the use of endophytes in agriculture. This will involve testing different endophyte inoculation methods, optimizing their efficacy, analysing human impacts, and developing strategies to scale-up their use in farming systems. It is also important to assess the potential risks associated with the use of endophytes, such as the spread of harmful microbes or impacts on soil microbiota.

Due to its novelty future, endophyte studies can take on a range of scopes, whether it be the development of stress tolerant plants, bio inoculums, natural antimicrobials/pesticides, or understanding the fundamental impacts of stress on, plants, the endomicrobiome, and their relationship. There are many contributions still required to realise the potential of endophytes in agriculture. Any and all work conducted on endophytes helps fill in the gaps in the knowledge of endophytes and their properties to one day have enough data and evidence to make a broader and clearer analysis of endophyte-based approaches to agriculture.

Overall, the results presented in this review paper demonstrate the immense potential of endophytes in agriculture and highlight the need for further research and development in this field. By harnessing the benefits of endophytes, we can move towards a more sustainable and efficient agricultural system, promoting both environmental and economic sustainability, all the while, fortifying crop plants against stressors and reducing the need for chemical treatments of the land.

## Figures and Tables

**Figure 1 microorganisms-11-01276-f001:**
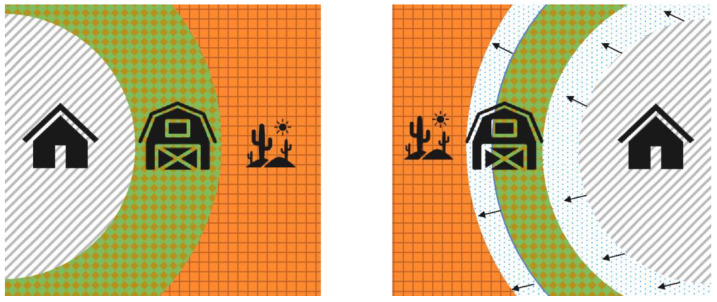
Visual representation of the effects of an increasing population requiring more housing. Increasing the demand for housing space and township expansion encroaches on farmland, some of which is unable to move relative to this growth. An increased efficiency is required from reduced farmland or crops will have to adapt to survive in these marginal conditions. Endophytes can offer potential solutions for the aforementioned issues.

**Figure 3 microorganisms-11-01276-f003:**
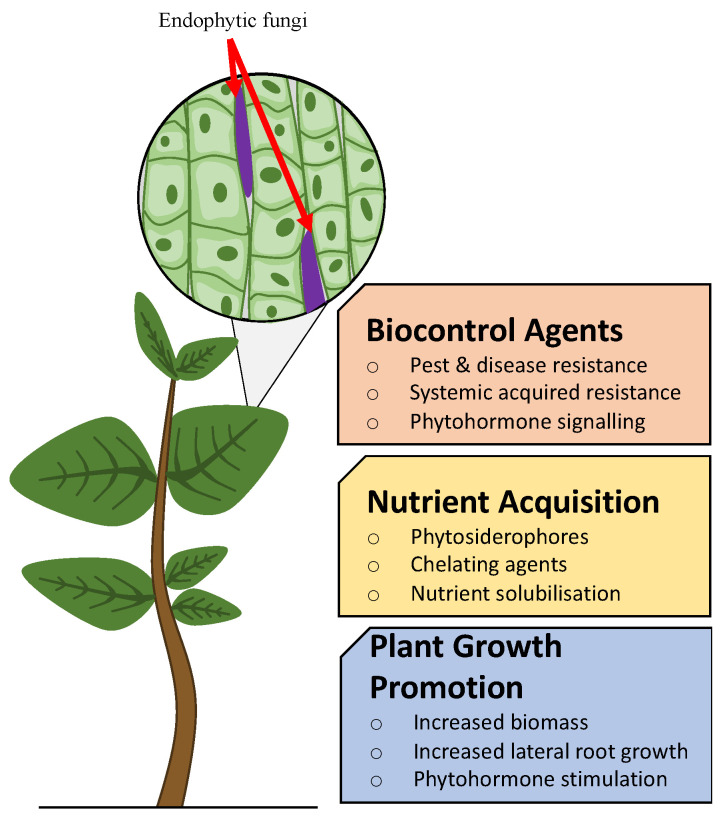
Overview of range of applications and desirable endophytic impacts upon host plants.

**Figure 4 microorganisms-11-01276-f004:**
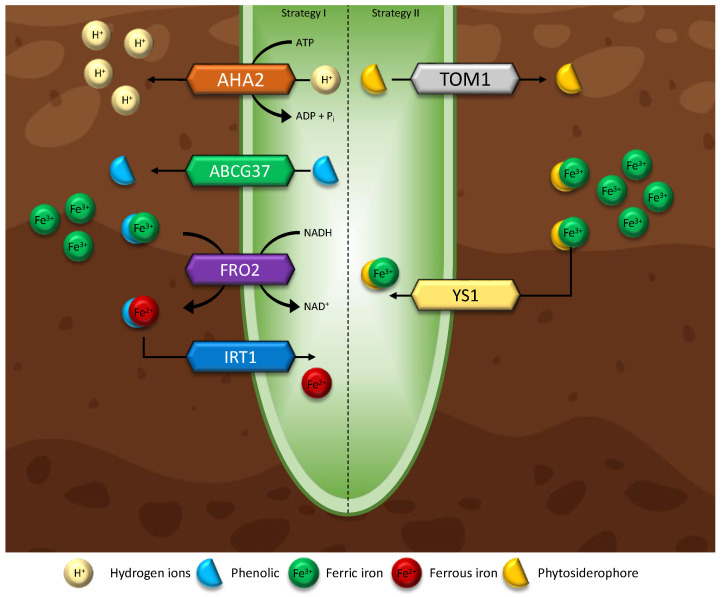
Iron accumulation strategies. (**left**) Strategy I is conducted by most non-grass plants, whereby iron is taken into the plant’s epidermis through a reduction of ferric iron to ferrous iron (FRO2). This pathway involves H^+^ soil acidification via ATPase AHA2 and phenolic excretion to improve iron availability. (**Right**) strategy II plants are mainly grasses and can directly uptake ferric iron when complexed with phytosiderophore (DMA) through transporters TOM1 and YS1. This pathway is much more energy efficient and tends to have a higher rate of iron accumulation.

**Figure 5 microorganisms-11-01276-f005:**
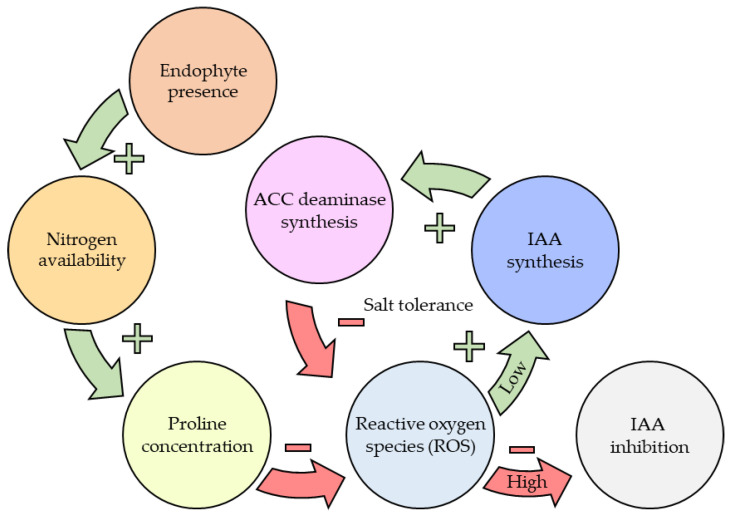
Flowchart outlining the flow on effect that basic nutritional improvements, such as nitrogen, have on downstream processes. Endophyte association can improve a plants ability to metabolise ROS, reducing accumulation and resulting in a greater tolerance to saline conditions.

**Figure 6 microorganisms-11-01276-f006:**
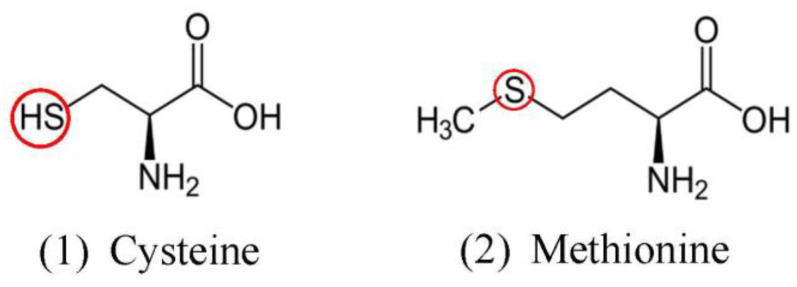
Sulphur containing amino acids found in plant proteins.

**Table 2 microorganisms-11-01276-t002:** Summary table of key plant signalling molecules produced, stimulated, or co-produced by endophytes and their effects on plant health.

Signalling Molecule	Effect	Reference
Auxins (IAA)	Promote cell elongation, root development, and apical dominance.	[83,90,149,191,192]
Gibberellins (GA)	Stem elongation, flowering, and leaf spread.	[48,49,150,159,193]
Abscisic acid (ABA)	Promote cellular conservation of water under drought and salinity stress.	[83,90,193,194]
Salicylic acid (SA)	Induces SAR combatting pathogen infection.	[78,160,161,166,168,169,177]
Jasmonic acid (JA)	Defence against pests through deterrence and elimination of pests.	[66,141,142,160,162]
Cytokinin’s (CK)	Promotes cell division, apical dominance, lateral root growth.	[151,176,177,178,179,180,181,182]
Reactive oxygen species (ROS)	Involved in stress signalling and programmed cell death. Major issues with accumulation resulting in toxification.	[104,133,140,184,186,187,195,196]

## Data Availability

No new data were created or analyzed in this study. Data sharing is not applicable to this article.

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
