# Peer review of "Endophytes in Agriculture: Potential to Improve Yields and Tolerances of Agricultural Crops"

_microorganisms, 2023, doi:10.3390/microorganisms11051276_

Round 1

Reviewer 1 Report

The manuscript on Endophytes in agriculture: potential to improve yields and tolerances of agroeconomical crops submitted by Watts et al. summarized the current knowledge on endophytes in agriculture, highlighting the potential as a sustainable solution for improving crop productivity and general plant health. The topic of the manuscript does fit the scope of the journal well. The paper is clearly structured and well organized. But there are some problems need to be solved. I marked them on the attached file. In addition, the styles of the text and references should be checked carefully.

Author Response

Dear reviewer, we wish to thank you for your contribution to the paper. Please find attached file with response to feedback. Regards.

Reviewer 2 Report

Review entitled “Endophytes in Agriculture: Potential to Improve Yields and Tolerances of Agroeconomical Crops” by Watts et al. reported the potential applications of endophytic microbes, bacteria and fungi in agricultural sectors. The review contains high-impact data and there are some minor clarifications before being accepted to be published in the microorganism journal.

1-    To increase the visibility of the review, I recommend adding the list of content after the Keywords.

2-    Please refer to the figures in text before their cited, for instance, Figure 1 does not refer to it in the text. Please check and revise the manuscript.

3-    Figure 1 needs more explanation or remove it.

4-    Figure 2 containing A and B, please add “A” and “B” on the figure. This comment will apply to all figures containing more than one panel.

5-    I recommend separating Table 1 to two tables, one for fungi and one for bacteria. Also, citing more types of endophytic fungi and bacteria have potential roles in agriculture. I recommend the following for citing:

https://doi.org/10.1016/j.aoas.2015.04.001; https://doi.org/10.3390/plants10010076; https://doi.org/10.3390/biom11020140.

6-    The authors neglected one important type of endophytic organism, actinomycetes, what about this type as endophytes?

7-    Data in sections 5, 6, and 7 can be summarized in tables.

8-    Sections 5, 6, and 7 should to be subsections under section “Role of endophytes in plant health”

9-    The discussion of efficacy of endophytes to improve plant growth via secretion of phytohormones can be improved using the following references: https://doi.org/10.3390/cells10051059; https://doi.org/10.1515/bmc-2021-0019; https://doi.org/10.3390/plants10050935.

10- The scientific names of plant and organisms must be in italics, please revise throughout the manuscript.

11- Conclusion should be rephrased to refer to the prospective study.

The review needs moderate editing of English language 

Author Response

(The authors gave the same response as above.)

Reviewer 3 Report

The article “Endophytes in Agriculture: Potential to Improve Yields and Tolerances of Agroeconomical Crops ” is devoted to the important and acute theme of the role of endophytic microorganisms in modern agriculture. The authors provided a lot of information on endophytes properties and give some characteristics of the situation in which agriculture is set up now. There are some comments:

Whole text: Improve references (Delete families in references like this “Khan, Hussain, Al-Harrasi, Al-Rawahi and Lee [149]”, see guide for authors.). Include strain definitions then it possible (not B. subtilis, but B. subtilis 26D for example)

Title:  there is no such thing as Agroeconomical Crops. You can change it on “Agricultural Crops”. If you would like to say about agroeconomy, please, provide data on world market of biopesticides, for example.

1)Manuscript  should be built from General to specific. Paragraph "2. Food supply insecurity: climate change and population growth" must be the first. Food insecurity is the problem, endophytic microbes using it is one of the ways to dissolve the problem, isn’t it? This paragraph, however, should be improved and become more realistic. Situations in Africa demand fertilizers and irrigation, not fine-tuning systems using endophytic microbes. European farmers, for example, are faced with the problem of limitation of chemical fertilizers and pesticides use, and in this situation biofertilizers can be the "lifeline". Please, collect data on current situations in the world. It is not necessary to give a divided paragraph "2.1 Crop loss". 

The paragraph 3. Agricultural potential of endophytes must be shortened and essential information (for example, the definition of the term “endomicrobiome”) must be included in the section which now is Introduction. 

Authors should provide information on methods of providing evidence of endophytic properties of microbes, it is a very important problem now.

Table 1 must be improved. Firstly, this table is about endophytes, why does it contain N/A plant hosts? Some endophytic microbes are distributed world-wide and in a wide range of plant hosts. For example, B. subtilis is cosmopolitan, one of/the most common endophytes along with B. megaterium, Pseudomonas sp and B. thuringiensis, but in the table it has N/A host and Vigna radiata host. It gives the false conception of endophytes presence in plants.

"4.Plant Endophyte Relationship" must be shortened and an essential information on plant-microbe interaction must be added. Now it is a fragment of text with repetitive points. Besides "production of phytohormones", it is an important part of plant-microbe relationships. 

5.Paragraph on nutrient uptake is informative, but figure 5 gives a reductionistic scheme of the influence of endophytes on drought tolerance. I advise you to delete it.

Delete 9. Amalgamating Traditional Knowledge with Modern Science.

10.Conclusions. You must concretize your own conclusions, emanating from paragraphs of the manuscript.  How do you see solution of the main problem, postulated in the article using endophytes?

Minor editing of English language required

Author Response

(The authors gave the same response as above.)

Reviewer 4 Report

Dear Authors! Make some changes:

1) What phytohormones are stimulated (Figure 3)? Is there an accumulation of plant growth-stimulating or resistance-inducing phytohormones?

2) Specify which hormones are produced (Table 1, page 7).

3) I did not understand the meaning of the sentences: "Potentially outlining ... relationship"(page 8), "Whereby EF are able ... was measured" (page 8), "well Furthermore, these strains ... respectively" (page 10), "... deficiencies apply to more... across the board" (page 14), "Highlighting the overall complexity ..." (page 15). Please, rephrase them.

4) Give the references after the sentence: "Plants may have ... environments" (page 8).

5) Who releases plant hormones: plants or microbes? Specify this in the sentence: "Additionally, amongst ... nutrient uptake" (page 9).

6) Is there any information about the effect of pore forming toxic proteins on the human gastrointestinal tract (page 10)? Please, give the reference. 

7) ABA involved not only in biotic stress response of plants (page 11).

8) What kind of stress is mentioned in sentence: "Suggesting that endophytes are capable..." (page 18).

My recommendations for the design of the review:

1) Do not mention Prof., Dr. according instructions for authors (Title).

2) Align the paragraphs for width  (page 3, 11).

3) Check the spelling Epichloe (Table 1, page5).

4) Write Vigna radiata (Table 1, page 7), Graminaceous (page 13), and latin names in references in italics.

5) Don`t list all the authors (page 8, 11, 16, 18).

6) Check punctuation marks in the sentences: "It could be considered that ... to respond to stressors" (page 8), "Thus, identifying ... across the board" (page 14), "... whether appropriate regulators are involved" (page 15), " A study conducted by Varga" (page 16), "...in-vitro..." (page 17).

7) Add researchers in the sentence: "Many state that the SAR..." (page 12), Pseudomonas fluorescens - in " Endophyte G20-18 ... bacterial pathogen" (page 12), around the roots of - in "Some endophytes produce and excrete PS into ..." (page 13), Rahnella sp. and Burkholderia sp. - in "A study conducted by Varga ..." (page 16).

8) Write Nickel in small letter (page 14). Write poplus in capital letter (page 16).

9) Decipher for the first time EDTA, PAA, etc (page 15), Conc. (Figure 5).

10) You should short the names of the journals uniformly (References). Write the references according instructions for authors. For example, references number 150, 188.

Author Response

(The authors gave the same response as above.)

Round 2

Reviewer 3 Report

Dear authors,

I have received more or less satisfactory answers on my comments, except for Table 1.

1) Please insert strain definitions (for example, change B. subtilis to B. subtilis 26D etc) according to the literature source.

2) Remove or find the information about the plant host of each strain, or give other evidence of endophytic properties of each strain.

Now the table gives the false conception that, for example, all strains of B. subtilis can endophytically colonize only Vigna. In particular attention should be the strain of Botrytis cinerea, since generally it is a pathogen.

Author Response

Dear reviewer,

Thank you for your response, we have made the appropriate changes to strain information as per your first point. As for the second point, the host plant column lists the plant that the isolates originated from, not the effected/ treatment plant. The organisms that have 'not reported' alongside them do not have the source reported in the relevant reference, although each of them prove endophytic properties in their application. We have altered the header of this column to avoid any further confusion.